# Harnessing the Web and Knowledge Graphs for Automated Impact Investing Scoring

Qingzhi Hu*
University of Amsterdam
Amsterdam, The Netherlands
q.hu.catherine@gmail.com

Daniel Daza
Vrije Universiteit Amsterdam
University of Amsterdam
Discovery Lab, Elsevier
Amsterdam, The Netherlands
d.dazacruz@vu.nl

Laurens Swinkels
Erasmus University Rotterdam
Robeco Institutional Asset
Management
Rotterdam, The Netherlands
l.swinkels@robeco.com

Kristina Ūsaitė
Robeco Institutional Asset
Management
Rotterdam, The Netherlands
k.usaite@robeco.com

Robbert-Jan 't Hoen
Robeco Institutional Asset
Management
Rotterdam, The Netherlands
r.t.hoen@robeco.com

Paul Groth
University of Amsterdam
Discovery Lab, Elsevier
Amsterdam, The Netherlands
p.groth@uva.nl

## ABSTRACT

The Sustainable Development Goals (SDGs) were introduced by the United Nations in order to encourage policies and activities that help guarantee human prosperity and sustainability. SDG frameworks produced in the finance industry are designed to provide scores that indicate how well a company aligns with each of the 17 SDGs. This scoring enables a consistent assessment of investments that have the potential of building an inclusive and sustainable economy. As a result of the high quality and reliability required by such frameworks, the process of creating and maintaining them is time-consuming and requires extensive domain expertise. In this work, we describe a data-driven system that seeks to automate the process of creating an SDG framework. First, we propose a novel method for collecting and filtering a dataset of texts from different web sources and a knowledge graph relevant to a set of companies. We then implement and deploy classifiers trained with this data for predicting scores of alignment with SDGs for a given company. Our results indicate that our best performing model can accurately predict SDG scores with a micro average F1 score of 0.89, demonstrating the effectiveness of the proposed solution. We further describe how the integration of the models for its use by humans can be facilitated by providing explanations in the form of data relevant to a predicted score. We find that our proposed solution enables access to a large amount of information that analysts would normally not be able to process, resulting in an accurate prediction of SDG scores at a fraction of the cost.

## CCS CONCEPTS

• **Information systems** → **Data extraction and integration**; **Information systems applications**; **Information retrieval**; • **Applied computing** → **Law, social and behavioral sciences**; • **Networks** → *Network economics*.

## KEYWORDS

AI for sustainability, sustainable development goals, impact investing, web data mining, knowledge graph, graph neural networks, explainability

---

*Work done during internship at Robeco.

# 1 INTRODUCTION

In 2015, the United Nations introduced the Sustainable Development Goals (SDGs) [2], a set of 17 objectives proposed to governments and companies around the world aimed towards increasing human prosperity and sustainability. They address issues such as poverty, food access, health, education, gender equality, working conditions, climate change, energy, and life on land and water.

In the financial industry, SDGs serve as a guideline to determine suitable metrics for creating sustainable investment solutions. This requires the establishment of *frameworks* for measuring the impact that a particular investment can have on SDGs [31]. An SDG framework allows one to analyze a financial asset and produce an overall score of its alignment with the SDGs, enabling the creation of investment portfolios that have a positive impact on society and the environment.

At Robeco[1], we introduced the Robeco SDG score. With a framework that analyzes the impact of companies on the SDGs through the products they sell, the way they operate, and whether they are involved in controversies, companies receive an SDG score that indicates a negative, neutral, or positive alignment.

There are several challenges associated with producing SDG scores. First, substantial human effort is necessary to manually investigate reports, news, and other resources to unearth helpful information for evaluating how sustainable a company might be. Thus, the process is time-consuming and expensive, since expert knowledge is required to judge the relevance of different sources. Second, the information sources for assessing the activities of a company can often be fragmented, missing, or biased. For instance, only some businesses publish sustainability reports on their official websites. Even when available, they might be subject to greenwashing, which occurs when a company overstates the positive environmental impact of their activities.

Previous research has found that artificial intelligence (AI) has the potential to assist in meeting 79% of the targets of the SDGs [34]. Further work has shown different uses of AI for analyzing climate-related texts and financial disclosures [3, 5, 21]. Motivated by the success of machine learning in tackling other challenging problems

---

[1]https://www.robeco.com/

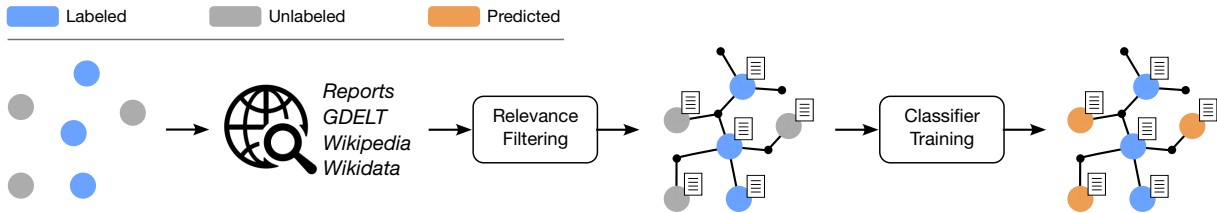

**Figure 1: Illustration of our pipeline for predicting SDG alignment scores. On the left, we start with a set of companies, some of which are labeled with SDG scores. We crawl the web for different textual and knowledge graph sources and filter them for relevance. The resulting data is used to train classifiers, which allow us to predict labels for unlabeled companies.**

that involve heterogeneous data sources, we aim to employ it to help ease the burden of creating an accurate, scalable, and reliable SDG framework.

In this work, we describe a system for automating obtaining SDG scores for a given set of companies. In particular,

- We outline the web data sources we collected, comprising sustainability reports, news, Wikipedia descriptions, and a subset of the Wikidata knowledge graph (KG).
- We describe a novel method for filtering and aggregating these sources into relevant features, that we use for training multiple classifiers that predict SDG alignment scores.
- We discuss the performance of the classifiers in predicting SDG scores and its impact on an explainable application towards automating the process of creating an SDG framework.

Our results show that the system can effectively extract the relevant data such that the classifiers *predict net alignment SDG scores with an average micro F1 score of 0.89*, demonstrating the effectiveness of our proposed solution. We find that textual features are responsible for the performance of the classifiers, while information from the graph is only beneficial when it is condensed. Lastly, we describe how we can explain predictions provided by the classifiers, such that experts are better informed when using them as a guide for generating SDG scores, and we provide a discussion on the implications of our results.

## 2 RELATED WORK

*SDG-related data sources.* Previous works have advocated for the creation of datasets that could be used in research on AI for SDG-aligned activities [9, 11, 22]. These works focus on specific activities and documents that describe activities in different sectors of the economy. In our work, we consider a broader set of sources obtained from the Web, that range from sustainability reports posted on company websites, news, Wikipedia descriptions, and a subset of the Wikidata KG[35]. This allows us to take into account a variety of factors to assess the impact of an organization on different SDGs.

Furthermore, recent works obtain labels for training machine learning models with automatic methods, such as weak supervision, instead of using gold labels from experts [9, 22]. This might lead to inconsistent SDG scores and may hinder the development of a transparent and coherent SDG framework [4]. We address this challenge by using SDG scores produced by experts in the domain.

*Text-based methods for estimating SDG alignment.* Previous work has considered the application of natural language processing (NLP) techniques to evaluate the impact of a company on SDGs [8]. The types of methods can be categorized into i) keyword-based approaches, ii) machine learning approaches based on hand-crafted features, and iii) machine learning approaches that incorporate context and end-to-end learned features [15]. The latter category has gained popularity due to the outstanding performance of large pre-trained language models like BERT and GPT-3 on language-related tasks [7, 10]. Such models have been extended to the sustainability domain, as in the case of ClimateBERT [36], which was pre-trained on a corpus of climate-related text to identify sections in sustainability reports relevant to climate change.

*Knowledge graphs for SDGs.* Network effects have been studied in the field of finance [17, 33], in areas such as portfolio diversification [20], portfolio selection [24], and the design of food supply chain networks to help achieve SDG goals [14]. These studies motivate incorporating domain specific networks, such as KGs [13] containing information about companies, their activities, and stakeholders, among others, into SDG frameworks.

Previous work has proposed a set of tools to aggregate and link multiple resources relevant to the SDGs [23] using Semantic Web technologies [1], in some cases also considering the effect of different countries towards SDGs[29]. In this work, we focus on the specific case of companies, and in addition to collecting a relevant KG, we use it as an additional source of information in a machine learning model that facilitates the estimation of SDG alignment.

## 3 SYSTEM DESCRIPTION

We address the problem of predicting how well aligned a company is to the SDGs, by using a collection of relevant data sources and labels produced by experts, and training a machine learning model to automate the task. More formally, we are given a set $C$ of companies, which is composed by two disjoint subsets $C_L$ and $C_U$ of labeled and unlabeled companies, respectively. Each of the companies in $C_L$ is accompanied by a sequence of SDG scores $(s_1, \ldots, s_{17})$, where each $s_i$ is a number that denotes how well-aligned a company is with each of the 17 SDGs. The goal consists of training a model to predict SDG scores for companies in $C_U$.

To achieve this, we build a pipeline that consists of data collection, relevance filtering, and model training and evaluation, as illustrated in Fig. 1, which we describe next.

## 3.1 Data Sources

*SDG scores.* We employ the data in the Robeco SDG framework and extract a subset of 1,391 companies that have been scored by expert analysts. The score is an integer that can take values between -3 and 3, indicating *strongly misaligned* and *strongly aligned*, respectively. For this subset we did not have access to enough training data related to SDGs 4, 10, and 17, which we omit in our experiments.

*Sustainability reports.* In order to assess how the products of a company might contribute to each of the SDGs, we scrape the Web in search for official sustainability reports. We make use of Bing's Web Search API [30] to construct queries that first search for the official domain of a company, and then for sustainability reports within that domain. From the resulting web pages, we extract any reports available in PDF format, and if no PDF file is found, we use the content of the web page.

*Wikipedia descriptions and news.* Relying on sustainability reports alone to evaluate the operations of a company might lead to a biased estimation of its impact on SDGs, due to issues such as green washing. To incorporate more diverse sources, we turn our attention to Wikipedia pages and news. Similar to sustainability reports, we search for a Wikipedia page associated with each company, and use the text in it as a source describing its operations. We then use the Wikipedia pages to link companies to entities in the GDELT database [18]. GDELT is an open-source project that analyzes news media in over 100 languages, in print, broadcast, and online formats. Since every entity in the GDELT database is associated with a Wikipedia URL, we use it to retrieve any news from 2021 associated with the companies of our interest.

*Knowledge Graphs.* In order to leverage the structural relationships that exist between companies, we map each of them to an entity in the Wikidata KG [35]. We then retrieve a subgraph around the companies, by iteratively extracting neighboring nodes in the graph until all companies are reachable with each other within 4 edges. The resulting subgraph contains 74,840 nodes, 160,994 edges, and 610 different relation types. Examples of the relation types that we extract are *country, headquartes location, subsidiary, language used, owner of, industry,* and *owned by*.

We additionally construct a summary of the KG: an undirected graph where nodes are companies and an edge exists between them if they are within two steps in the original KG.

## 3.2 Relevance filtering

The dataset collected with the previous section provides different views from which to assess the impact of a company on SDGs. However, it also generates a vast amount of text that would increase the cost of model training and hyperparameter optimization, while containing several instances that are not relevant for the task. To select relevant sentences from Wikipedia pages, for each SDG, we collect keywords related to them and concatenate them to form a sentence. We then use SBERT [25] to map this sentence into a single embedding. We also use SBERT to embed sentences from the Wikipedia pages, and then select the top 5 sentences with highest similarity with the embedded keywords. To increase precision, we follow this by BERT-NLI [12], a natural language inference model that we use to detect if a sentence is entailed from the description

**Table 1: Classification results, measured by the micro and macro F1 score for the three models in our experiments.**

| SDG | Micro F1 | | | Macro F1 | | |
|---|---|---|---|---|---|---|
| | BRF | R-GCN | GCN | BRF | R-GCN | GCN |
| 1 | 0.92 | 0.87 | 0.92 | 0.30 | 0.26 | 0.26 |
| 2 | 0.95 | 0.88 | 0.95 | 0.16 | 0.15 | 0.16 |
| 3 | 0.83 | 0.72 | 0.81 | 0.25 | 0.22 | 0.22 |
| 5 | 0.97 | 0.93 | 0.97 | 0.16 | 0.16 | 0.25 |
| 6 | 0.97 | 0.93 | 0.97 | 0.19 | 0.19 | 0.24 |
| 7 | 0.86 | 0.77 | 0.85 | 0.15 | 0.15 | 0.15 |
| 8 | 0.77 | 0.64 | 0.71 | 0.22 | 0.22 | 0.31 |
| 9 | 0.65 | 0.59 | 0.65 | 0.17 | 0.17 | 0.25 |
| 11 | 0.80 | 0.75 | 0.81 | 0.20 | 0.20 | 0.18 |
| 12 | 0.91 | 0.84 | 0.91 | 0.21 | 0.21 | 0.24 |
| 13 | 0.90 | 0.85 | 0.90 | 0.20 | 0.20 | 0.17 |
| 14 | 0.98 | 0.96 | 0.99 | 0.20 | 0.20 | 0.30 |
| 15 | 0.96 | 0.89 | 0.96 | 0.18 | 0.18 | 0.16 |
| 16 | 0.95 | 0.93 | 0.95 | 0.18 | 0.18 | 0.16 |
| Average | **0.89** | 0.83 | 0.88 | 0.20 | 0.19 | **0.22** |

of an SDG. We provide some examples of the resulting data in Appendix A.

## 3.3 Models

We treat the problem of SDG score prediction as a classification task, where the labels correspond to each of the integer scores between -3 and 3. We leave more structured models such as ordinal regression for future work [32]. As features for training predictive models, we use bag-of-words (BOW) representations extracted from all our textual data, as well as the KG or the summary graph for models that can additionally process graph-structured data.

The first model we consider is a Balanced Random Forest (BRF) that predicts from BOW features only [19], since it is a computationally efficient model that can also deal with different class frequencies in our data.

As graph-based models, we consider the Graph Convolutional Network (GCN, [16]), which we train using the summary graph; and the Relational GCN (R-GCN, [27]) trained on the original KG. We train the models for 5,000 epochs with the Adam optimizer and learning rate of 0.01. Both models use two layers with a hidden size of 16, and ReLU activation functions.

## 4 RESULTS

To evaluate the effectiveness of the models, we analyze their accuracy at predicting SDG scores for a held-out set of companies. Additionally, we evaluate a case study where we provide explanations for predictions given by the model in the form of relevant terms in the input for providing a particular score.

## 4.1 Predictive performance

We measure the performance of the classifiers by computing the F1 score. Since the predictions range over multiple classes, we compute micro F1 scores (where true positives, false positives, and

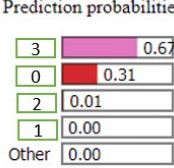
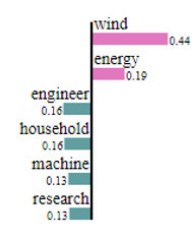
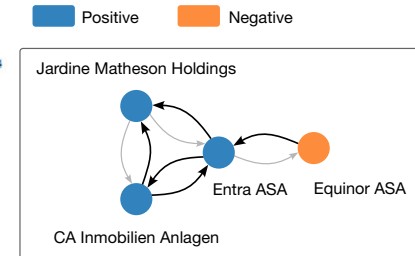

(a) Textual features

(b) Graph features

**Figure 2: Example of an explanation for predicting an alignment score for SDG 7 (affordable and sustainable energy), using textual features (Fig. 2a) and graph features (Fig. 2b).** *Fig. 2a*: **Given a bag-of-words representation of a company (left), the highest probability (center) is assigned to score 3 (strongly aligned), with the terms** *wind* **and** *energy* **being responsible for the prediction (right).** *Fig. 2b*: **We highlight the edges selected that according to a graph explanation model, are responsible for classifying the company** *Entra ASA* **as positively aligned with SDG7.**

false negatives are computed regardless of the class) and macro F1 scores (computed for each class separately, and then averaged). The results are shown in Table 1.

When inspecting the micro F1 scores, the highest value that we obtain is 0.89, corresponding to the BRF, closely followed by the GCN. This shows that the large amount of textual features that we collected is responsible for the ability to predict SDG scores, and adding the graph (as in the GCN model) does not bring significant improvements. In the case of the R-GCN, the KG can even yield lower performance. We attribute this to the fact that in the KG used by the R-GCN, only company nodes had textual features, whereas other nodes used learned embeddings that require additional training. This is demonstrated by the better performance of the GCN, where the graph consists exclusively of companies associated with textual features. We note that the macro F1 scores are much lower, since these treat all classes equally. In this case, the score decreases after being dominated by classes not commonly used by expert analysts, for which there is not sufficient training data.

## 4.2 Explainability

In addition to a model that is successful at predicting SDG alignment scores, we are interested in its deployment in an application for automated SDG scoring where humans can use predictions as a guide that could inform their reasoning and decision. This motivates the implementation of a mechanism for providing explanations for the predictions, such that they can be trusted and allow for the identification of potential biases or systematic errors before they occur during deployment.

To this end, we use LIME [26] to generate a score that indicates how relevant each of the terms in the BOW representation is responsible for a particular prediction. This is illustrated in Fig. 2 (a), where we explain a score predicted for SDG 7 (ensuring access to affordable, reliable, sustainable and modern energy for all) for the company Vestas Wind Systems. We observe that the model assigns the largest probability to a score of 3 (strongly aligned), which LIME attributes to the terms *wind* and *energy*.

Using a graph cluster algorithm [28], we cluster companies in the company graph into 50 clusters. Each company is assigned its SDG label, and the mean label for the cluster becomes the new

label for all companies within it. For companies without an initial SDG label, GCN is utilized for label classification. To explicate the classification results of the graph, we employ GNNExplainer [37], which reveals insights through two dimensions: the significant subgraph consisting of important neighbors and connections related to the focal node, and the influential feature driving the classification outcomes for that node. Fig. 2 (b) illustrates the pertinent subgraph explaining the prediction of Entra ASA. It indicates that Entra ASA is categorized in the same group as CA Immobilien Anlagen AG and Jardine Matheson Holdings Ltd, rather than Equinor ASA. Both CA Immobilien Anlagen AG and Equinor ASA play a direct role in determining the classification results for Entra ASA. CA Immobilien Anlagen AG and Entra ASA operate in the Real Estate Management and Development sectors with a neutral product score, while Equinor ASA belongs to the Oil, Gas, and Consumable Fuels sector with an extremely negative product score and operational (news sentiment) score.

The use of LIME enables term relevance visualization, offering insights into the factors influencing model predictions. This not only improves the transparency of our predictions, as exemplified in the Vestas Wind Systems case, but also facilitates the identification and mitigation of potential biases or systematic errors in our model. Hence, explainable AI techniques like LIME contribute to a more reliable and accountable AI-based SDG scoring process, promoting a nuanced understanding of a company's sustainability alignment and enhancing the robustness of the overall system. Based on these promising qualitative findings, in future work we plan to carry out a rigorous evaluation that quantifies how useful the explanations are to domain experts.

## 5 DISCUSSION

The opportunity set for Robeco's investment products consists of several thousands of companies in global developed and emerging markets, which is too large for analysts who have to continuously evaluate each company on their SDG alignment. Therefore, we have been developing SDG scores using AI methods to significantly increase the number of companies with SDG score coverage. This automated SDG scoring uses numerous data sources, including text data, to evaluate the impact of a company's products on SDGs

and whether they engage in controversial behavior. In addition, the automated SDG scores can be compared to the opinion of the domain experts, who can incorporate new insights provided by the automated SDG scores in their assessment.

This project has explored an extended set of publicly available sustainability data for creating SDG scores. The automated analysis has not only made it more efficient to transform the vast amount of (text) data into insights, but also unlock access to information that analysts normally would not be able to easily process themselves, such as news in multiple languages or complex relationships. Even for the most cases where we use the final verdict of the domain expert, the additional data and the quality of processing it and linking it through the network leads to a better and easier explainable SDG score at a fraction of the cost.

We see two possible directions for further improvement. First, we may want to explicitly search and adjust for possible corporate greenwashing [6]. Second, most corporate reporting and news deals with past corporate behavior, but forward-looking measures that predict future corporate behavior are even more relevant.

## 6  CONCLUSION

In this work, we presented a data driven procedure for tackling real-world business challenges in the sphere of impact investing. We demonstrate the efficacy of utilizing web data to address data scarcity issues for sustainability ratings. In addition, we demonstrate how contemporary NLP approaches can effectively identify important SDG objectives in a huge volume of unstructured content. We validate the use of our dataset by predicting the existing SDG scores developed by the investing firm Robeco, achieving a high micro F1 score performance, and we explore a method for explaining prediction in a way that expert analysts can interpret. In future work we would like to explore improved methods for detecting behavior such as green washing, as well as more expressive models such as language models while preserving explainability.

## ACKNOWLEDGMENTS

This project was partially funded by Elsevier's Discovery Lab.

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

# A DATA SAMPLES

In this appendix we provide examples of the texts we collected using sustainability reports and websites (Table 2), and Wikipedia descriptions (Table 3).

**Table 2: Sample of sentences collected from sustainability reports and websites related to SDG 13 (climate action).**

| Company | Resource | Summarized key actions |
|---|---|---|
| Cleanaway Waste Management Ltd | web | by reducing our greenhouse gas emissions , by the responsible management of our landfill gas , and by assisting our customers and the community in managing their waste impacts |
| Singapore Telecommunications Ltd | report | undertook a science based targets programme and engaged experts on developing science based targets |
| GlaxoSmithKline PLC | web | our climate strategy covers the full value chain of emissions reductions |
| Mapfre SA | web | protects the environment through public commitments |
| Carrefour SA | report | structured its climate action plan around three priority areas |
| Swiss Re AG | report | we use our existing processes and instruments to address climate - relat |
| Solvay SA | web | raising the bar |
| Crown Holdings Inc | web | drive climate action throughout our value chain |
| Enagas SA | report | through efficient use of energy |
| Cie Generale des Etablissements Michelin SCA | report | taking action both downstream from its operations to ght climate change , conserve natural protect objectives for 2050 to make all the production plants , supply chain operations and raw material and component inputs carbon neutral |
| NextEra Energy Inc | report | prepare our business to adapt to the effects of climate change |

**Table 3: Sample of sentences collected from Wikipedia after filtering.**

| Company | Product information |
|---|---|
| SGL Carbon SE | It is one of the worlds leading manufacturers of products from 29 production sites around the globe (16 in Europe, 8 in North America and 5 in Asia), and a service network in over 100 countries, SGL Carbon is a globally operating company |
| Gerdau SA | These products are used in different sectors, such as industry, metallurgy, farming and livestock, civil construction, automotive industries, petrochemicals, railway and naval sectors, in addition to orthodontic, medical and food areas |
| Bridgestone Corp | Today, Bridgestone diversified operations encompass automotive components, industrial products, polyurethane foam products, construction materials, parts and materials for electronic equipment, bicycles and sporting goods |
| MTS Systems Corp | The companys products and services support customers in research and development and QAQC testing of products through the physical characterization of materials, such as ceramics, composites and steel |
| Berkshire Hathaway Inc | Moore formulates, manufactures, and sells architectural coatings that are available primarily in the United States and 2001, Berkshire acquired three additional building products companies |
| ANDRITZ AG | Xerium Technologies is a global manufacturer and supplier of machine clothing (forming fabrics, press felts, drying fabrics) and roll covers for paper, tissue, and board machines |
| 3D Systems Corp | Applications and industries 3D Systems products and services are used across industries to assist, either in part or in full, the design, manufacture andor marketing processes |
| International Paper Co | At the time of sale, Temple-Inlands corrugated packaging operation consisted of 7 mills and 59 converting facilities as well as the building products operation |