# OpenReview forum: "Harnessing the Web and Knowledge Graphs for Automated Impact Investing Scoring"
_KDD.org/2023/Workshop/Fragile_Earth — KDD 2023 Workshop Fragile Earth Submission_

### Official Review · Reviewer_dK2p · 2023-07-10
**Problem is well posed and application of AI to automate the task of calculating assessment metrics of alignment of companies to their sustainability goals is provided. The paper is somewhat lackluster on technical details. However , it provides performance metrics and compares it different methods and hence gets my acceptance.**

**Rating:** 6
**Confidence:** 2

**Review:**

- Summary: Paper provides a method to automate the process of giving qualitative scores to corporations on their Sustainable Development Goals. On text and knowledge graph based data sets Natural Language Processing has been applied to perform classification task to give a qualitative score
- Strengths:

     - a.  Problem is described well and related to application of AI for quality assessment of how well aligned companies are to the
                     sustainable development goals.
     - b. Performance Metrics have been giving and evaluated against other methods.

- Weaknesses:

    - a. Details about the methods applied is not clearly explained.

    - b. No details regarding hyper-parameter tuning is provided.



- Questions: What happens if Key words used to learn embeddings are learnt by companies to make Green washing more robust to such prediction tasks?
- Limitations: Even though Green washing has been discussed, it still is a major limitation that needs more consideration .

---

### Official Review · Reviewer_F8No · 2023-07-16
**Review for "Harnessing the Web and Knowledge Graphs for Automated Impact Investing Scoring"**

**Rating:** 7
**Confidence:** 3

**Review:**

Summary:

In this paper, the authors propose a novel method, i.e., collecting and filtering a dataset of texts from different web sources and a knowledge graph relevant to a set of companies. The experimental results demonstrate that the effectiveness of the proposed solution.

Strengths:
- It is a genuinely exciting paper and this paper is generally well-written and easy to follow.
- Experimental results are promising and explainability is very helpful.

Weaknesses:
- For explainability, can the authors also compare it with the runner-up?

---

### Decision · Program_Chairs · 2023-07-19

**Decision:**

Accept (Oral)

**Comment:**

Congratulations!

We are pleased to inform you that your submission: Harnessing the Web and Knowledge Graphs for Automated Impact Investing Scoring has been accepted to The KDD 2023 Workshop Fragile Earth: AI for Climate Sustainability - from Wildfire Disaster Management to Public Health and Beyond.

Camera ready deadline is ** July 24 AOE **.  Please log in to OpenReview and prepare your camera-ready version based on the reviews. Formatting rules are the same as for the initial submission and submissions must adhere to KDD 2023 guidelines available at https://authors.acm.org/proceedings/production-information/taps-production-workflow.

Again, congratulations on the acceptance of your paper!  We look forward to seeing you at the workshop on Aug 7, 2023.

The Fragile Earth Workshop Proceeding Chairs